# Morphometric Analysis of Developmental Alterations in the Small Intestine of Goose

**DOI:** 10.3390/ani13203292

**Published:** 2023-10-21

**Authors:** Ligia Hiżewska, Cezary Osiak-Wicha, Ewa Tomaszewska, Siemowit Muszyński, Piotr Dobrowolski, Krzysztof Andres, Tomasz Schwarz, Marcin B. Arciszewski

**Affiliations:** 1Department of Animal Anatomy and Histology, Faculty of Veterinary Medicine, University of Life Sciences in Lublin, Akademicka 12, 20-950 Lublin, Poland; ligia.hizewska@up.lublin.pl (L.H.); cezary.wicha@up.lublin.pl (C.O.-W.); 2Department of Animal Physiology, Faculty of Veterinary Medicine, University of Life Sciences in Lublin, Akademicka 12, 20-950 Lublin, Poland; ewarst@interia.pl; 3Department of Biophysics, Faculty of Environmental Biology, University of Life Sciences in Lublin, Akademicka 13, 20-950 Lublin, Poland; siemowit.muszynski@up.lublin.pl; 4Department of Functional Anatomy and Cytobiology, Faculty of Biology and Biotechnology, Maria Curie-Sklodowska University, Akademicka St. 19, 20-033 Lublin, Poland; piotr.dobrowolski@umcs.lublin.pl; 5Department of Animal Reproduction, Anatomy and Genomics, University of Agriculture in Cracow, Mickiewicza Alley 24/28, 30-059 Cracow, Poland; krzysztof.andres@urk.edu.pl; 6Department of Animal Genetics, Breeding and Ethology, Faculty of Animal Sciences, University of Agriculture in Cracow, Mickiewicza Alley 24/28, 30-059 Cracow, Poland; tomasz.schwarz@urk.edu.pl

**Keywords:** ontogeny, duodenum, jejunum, liver, waterfowl, birds

## Abstract

**Simple Summary:**

This study concerned histological changes in the small intestine (duodenum and jejunum) and liver in the first weeks of life of geese. After analyzing the results, an increase in most traits was found during the development of the animals; the largest increase occurred during the period of changing the feed given to the geese. The research carried out on the digestive system of geese in the first weeks of life is important for zootechnics, animal breeding, and the food industry, helping to optimize breeding processes and improve the health and performance of these birds.

**Abstract:**

In this study, a morphometric analysis of morphological changes in the layers of the small intestine (duodenum and jejunum) and liver occurring during the hatching period (week 0) and postnatal development (weeks 1, 3, 6, and 8) was performed in geese. For this purpose, the staining of samples obtained from tissues collected from geese after culling was carried out. Staining was performed using the Goldner method to visualize all layers of the intestine for morphometric measurements. Our analysis focused mainly on traits such as the thickness of the mucosal, submucosal, and muscular layers, as well as traits related to intestinal absorption, such as the height and width of intestinal villi and crypts. Additionally, we also took into account the number of mononuclear and binucleate hepatocytes and other cells present in the liver. After analyzing the results, an increase in most traits was found during the development of the animals, with slight differences between the sections of the duodenum and jejunum. An interesting phenomenon was also noticed—the greatest increase in most traits was observed between the 3rd and 6th week of life, which coincides with the time of feed change. We hope that our work will highlight how important the digestive system is for birds because research on this topic is limited.

## 1. Introduction

The fetal development of birds is a long and complex process that varies between bird species. However, due to the immaturity of newly hatched birds, the post-hatching season may be critical for their survival and development. In the first weeks after hatching, significant changes occur to allow the survival of the growing organisms [1]. These processes involve almost all the crucial systems, but the maturation of the immune system, the skeletal system, and the digestive system are considered to be the most vital [2]. The rapid growth of the bird’s gastrointestinal system allows for increased feed intake and more efficient energy utilization, which is essential for the proper growth of birds [1].

Similar to mammals, the small intestine of birds is divided into three sections: duodenum, jejunum, and ileum [3]. The small intestine consists of longitudinal and circular musculature, which are responsible for mixing the food materials, as well as ridged submucosa and mucosa. The ridging of the mucosa comes from the need to increase the absorptive surface of the small intestine [4]. Circular folds formed within and along the submucosa, as well as the presence of numerous intestinal villi, serve to increase the surface area of absorption. The other characteristic structures of the small intestinal mucosa are the intestinal crypts, which are defined as a recess of the epithelium reaching the muscularis laminae of the mucosa [5]. The small intestine, particularly the villi and crypts, plays a key role in the absorption of nutrients and, consequently, plays a part in the subsequent stages of the digestion process. The epithelium of the mucous membrane consists of goblet cells, which are essential for maintaining intestinal homeostasis since they produce carbohydrates and glycoproteins [6]. The peptic digestion starts in the duodenum. In the jejunum, the main phase of the digestion process takes place, while the final processes of food absorption occur in the ileum [5].

The liver is a physiologically significant organ that participates in many crucial processes in the developing organisms. It is the central digestive gland that plays a crucial role in processing and storing nutrients. One of its key functions is the production of bile necessary for fat digestion, which is then delivered to the duodenum [7,8]. The liver also stores glucose in the form of glycogen, ensuring a constant energy supply to the body, as this is particularly vital during growth and development. Furthermore, the liver plays a significant role in detoxifying the organism by eliminating toxins and harmful substances. All of these processes are essential for the proper development and functioning of the organism during its youth [9,10]. Most birds’ diseases are associated with impaired liver function, and their specimens are commonly sent to the histopathological laboratory. A histopathological examination of liver samples is essential for the accurate diagnosis, monitoring, and management of various liver conditions [11]. The organ’s parenchyma consists of a uniform mass of hepatocytes, arranged in trabeculae that are typically two hepatocytes wide (in most bird species). Similar to mammals, it exhibits Disse spaces and stellate cells, sinusoids lined by Kupffer cells, and hepatic triads [12].

Several studies indicate that the size and digestive activity of the small intestine vary depending on the stage of growth [2,13,14,15]. However, the analysis of the literature shows that such studies mainly focus on mammals, leaving a considerable gap in research focusing on the post-hatching maturation of the digestive tract of birds, especially the small intestine and liver. The results of morphological and developmental studies are often fundamental to understanding the adjustments of dietary requirements of birds. In this work, we hypothesized that the first weeks of the goose’s life determine the stage of individual growth and affect the microscopic structure of the different layers of the small intestine and the morphology of the liver. We assumed that introducing a new feed into the goose diet may affect the variability of most characteristics in studied organs. Significant changes in the jejunum were also anticipated because it is where the primary absorption of nutrients occurs. To verify this, we investigated, through a morphometric analysis, the morphological changes in the layers of the small intestine (duodenum and jejunum) and also the liver that occur during postnatal development at weeks 0, 1, 3, 6, and 8 in geese.

## 2. Materials and Methods

### 2.1. Animal

The research was conducted on Zatorska geese, a breed indigenous to Poland and valued for its meat-producing traits. These geese are part of a dedicated waterfowl genetic conservation program. The birds were raised at the Educational and Research Hub affiliated with the Animal Science Department of Cracow’s Agricultural University. For the first three weeks, the young geese were kept indoors without pasture access. Beginning in their fourth week, they were relocated to a straw-bedded floor and given a minimum of eight hours of daily pasture time. The diet provided to the birds was meticulously designed to align with the national guidelines and best practices for poultry nutrition (Table 1). The study concluded at the end of week 8 because this is a critical time point for evaluating the health and development of geese. It aligns with national standard agricultural practices for assessing key metrics such as the growth rate and overall well-being [16] and is in accordance with the national genetic resource conservation program for goose populations [17].

### 2.2. Material

At week 0 post-hatch and, subsequently, at weeks 1, 3, 6, and 8 post-hatch, *n* = 6 male goslings were randomly selected from the flock. Each selected bird was weighed and then euthanized using cervical dislocation. For birds aged 3 weeks or older, this procedure was preceded by electric stunning (STZ-6, Koma, Bratislava, Slovakia). 

### 2.3. Tissue Samples Collection and Histomorphometrical Analysis

Duodenal (30 mm long; taken 20 mm distal to the pylorus) and jejunal segments (30 mm long; 50% of the total length of the jejunum) and samples of the right lobe of the liver were taken from each bird immediately after euthanasia. Dissected samples were fixed in 4% buffered formaldehyde (pH 7.0) for 24 h and washed with tap water. The samples were then dehydrated through a graded series of EtOH and fixed with the nonpolar solvents Ottix Plus and Ottix Sharper (DiaPath, Martinengo, Italy). Intestinal cross-sections were prepared by embedding them in paraffin, using the modular embedding machine (MYR EC-350, Casa Álvarez Material Científico SA, Madrid, Spain). Then, they were cut into 5 µm thick sections, using a rotary microtome (HM 360, Microm, Walldorf, Germany). These sections were then placed on SuperFrost^®^ Plus base slides (Thermo Scientific, Menzel-Glaser, Braunschweig, Germany) and dried on a hotplate heated to 48 °C. Goldner’s trichrome staining [19] was used to stain the sections, allowing for the visualization and histological differentiation of the layers of the examined sections of the intestinal wall. Photographic documentation was made using CX43 and BX63 light microscopes at the following magnifications: ×4, ×10, and ×60. The microscopic images obtained were then examined using the graphical analysis software Olympus cellSens (Olympus, Tokyo, Japan). The following traits were analyzed: the thickness of the inner and outer muscle layer, mucosa and submucosa thickness, crypt depth (defined as the depth of the invagination between adjacent villi, from the bottom of the crypt to the base of the villus) and width (measured in the middle of the crypt depth), villi height (from the tip of the villus to the villus–crypt junction) and width (measured in the middle of the villus height), total crypt number, villus-height-to-crypt-depth ratio (VH/CD ratio), number of villi per millimeter of mucosa, and number of enterocytes per 100 um of villus epithelium (Figure 1). The counts of mononuclear and binucleated hepatocytes, along with other cell types, were quantified in the liver. Histomorphometrical measurements were made with the use of graphic analysis software ImageJ 1.53 (National Institutes of Health, Bethesda, MD, USA, https://imagej.nih.gov/ij/index.html; accessed on 21 September 2023).

### 2.4. Data Analysis

From each analyzed intestinal segment, 10 measurements of each trait were taken. Utilizing the acquired measurements of intestinal segment structures, calculations were conducted to derive additional information concerning the proportions of distinct anatomical components. This analytical approach facilitated the quantification of the muscular layer’s thickness relative to the mucous layer, as well as the submucosa’s dimension in relation to the mucosal layer. Furthermore, an analysis was conducted to ascertain the absorptive surface area of the intestine. For this purpose, the methodology and equations delineated by Kisielinski, et al. [20] were employed. All of the results are presented as the mean value (with SD). Differences between means were assessed utilizing a two-way analysis of variance (ANOVA), followed by Tukey’s post hoc honest significant difference test to account for multiple comparisons. Two factors were considered in the study: the age of the geese and the intestine segment. Only consecutive time points were taken into account to track the progression of the examined traits pertaining to the intestine development. Student’s *t*-test was employed to assess intestine sections’ (duodenum and jejunum) differences at specific time intervals. One-way analysis of variance (ANOVA) was utilized to assess liver cell number changes between consecutive time points. The normal distribution of the data was confirmed using the W Shapiro–Wilk test, and the homogeneity of variance was examined using the Brown–Forsythe test. A two-tailed significance level (*p* value) lower than 0.05 was considered to indicate statistical significance. The statistical software GraphPad Prism version 9.5.1 for Windows (GraphPad Software, San Diego, CA, USA, www.graphpad.com; accessed on 21 September 2023) was utilized for all statistical analyses.

## 3. Results

In the period from hatching to the third week of life in geese, a significant increase in the thickness of the inner muscle layer was observed in both the duodenum and the jejunum (*p* < 0.001), which subsequently stabilized between the 3rd and 6th week of life. From the 6th to the 8th week of life, another significant increase in thickness was noted, primarily in the jejunum (*p* < 0.001) and, to a lesser extent, in the duodenum (*p* < 0.05; Figure 2A). When comparing the thickness of the inner muscle layer between both segments of the small intestine, it was found that the inner muscle layer of the jejunum was significantly thicker than that of the duodenum both at the 3rd week (*p* < 0.001) and at the 8th week (*p* < 0.05) of life (Figure 2A, Figure 3 and Figure 4).

In the thickness of the outer muscle layer, a significant increase was observed from hatching to the first week of life in both segments (*p* < 0.001), which then continued until the 3rd week in the case of the jejunum (*p* < 0.001) but stabilized in the duodenum. Between the 3rd and 6th week of life, the jejunum showed stabilization of the thickness of the outer muscle layer, while the duodenum exhibited slow growth (*p* < 0.001). In subsequent weeks, between the 6th and 8th weeks, a rapid increase in the thickness of the outer muscle layer was noted in both segments (*p* < 0.001; Figure 2B). In the 1st and 8th weeks of life, the outer muscle layer was significantly thicker in the duodenum (*p* < 0.001), while in the 3rd week, it was thicker in the jejunum (*p* < 0.001; Figure 2B).

Between hatching and the 1st week of life, a significant rapid decrease in the ratio of the thickness of the muscle layer to the mucosal layer was observed in both the duodenum and the jejunum (*p* < 0.001). Between the 1st and 3rd weeks of life, the jejunum experienced rapid growth (*p* < 0.001), while the duodenum stabilized. Between the 3rd and 6th weeks of life, the duodenum exhibited another, slower decrease in the ratio of muscle thickness to mucosal thickness (*p* < 0.05), while the jejunum remained stable, continuing until the end (Figure 2C). In the period of hatching and the 1st week, a difference was observed between the duodenum and the jejunum, with a greater ratio of muscle thickness to mucosal thickness in the duodenum (*p* < 0.05), whereas in the 6th and 8th weeks, the jejunum showed a significantly greater ratio (*p* < 0.001 and *p* < 0.05, respectively; Figure 2C).

A significant, rapid increase in the thickness of the mucosa in geese was observed between hatching and the 1st week of life in both sections of the small intestine (*p* < 0.001). Between the 1st and 3rd weeks of life, there was a stabilization in the thickness of the mucosa in the duodenum, while the increase progressed in the jejunum (*p* < 0.001) until the 3rd week. Afterward, there was a slow decrease (*p* < 0.01) in the jejunum and a significant increase in the duodenum until the 6th week (*p* < 0.001). Between the 6th and 8th weeks, there was an increase in the thickness of the mucosa in both sections (*p* < 0.001; Figure 2D). At the 3rd week, a significant difference was observed between the duodenum and the jejunum, with the thickness of the mucosa being greater in the jejunum (*p* < 0.001). However, at the 6th and 8th weeks, the thickness of the mucosa was greater in the duodenum (*p* < 0.001).

The submucosal thickness changed relatively evenly throughout the period in both segments of the intestine. Between hatching and the 3rd week of life, a significant increase was observed in the duodenum (*p* < 0.001), and a gradual increase was seen in the jejunum (*p* < 0.001). Then, between the 3rd and 6th weeks of life, a significant decrease in submucosal thickness occurred in both segments of the intestine (*p* < 0.001). Between the 6th and 8th weeks, another significant increase in submucosal thickness was observed in both segments (*p* < 0.001; Figure 2E). In the 1st week of life, there was a significant difference between the duodenum and the jejunum, with greater submucosal thickness in the jejunum (*p* < 0.001). However, by the 8th week, the submucosal thickness was greater in the duodenum (*p* < 0.001; Figure 2E).

Between hatching and the 1st week of life, a significant decrease in the ratio of submucosal thickness to mucosal thickness was observed in both the duodenum and the jejunum (*p* < 0.001). Between the 1st and 3rd weeks of life, the duodenum stabilized in submucosal thickness, while the jejunum showed an increase (*p* < 0.001). Between the 3rd and 6th weeks of life, there was a significant increase in the submucosal thickness in both segments of the intestine (*p* < 0.001), with a significant deepening between the 6th and 8th weeks in the jejunum (*p* < 0.001) and stabilization in the duodenum (Figure 2F). In the 1st and 6th weeks of life, there was a significant difference between the duodenum and the jejunum in the ratio of submucosal thickness to mucosal thickness, with the jejunum having a greater ratio (*p* < 0.001). In the 3rd week, the ratio was greater in the duodenum (*p* < 0.05; Figure 2F).

In both segments of the intestines, a significant increase in villus height was observed between hatching and the 1st week of life (*p* < 0.001). In the jejunum, this growth continued until the 8th week (*p* < 0.001, with an additional significant increase between the 6th and 8th weeks; *p* < 0.01), while in the duodenum, the growth only continued between the 3rd and 6th weeks of life (*p* < 0.001), with the remaining weeks maintaining a stable level (Figure 5A). The difference between both segments was observed only in the 6th and 8th weeks of life, where the height of the villi was significantly greater in the jejunum (*p* < 0.001; Figure 5A).

The results of the developmental changes in villus width revealed nearly identical trends to their height. The only deviation from this trend was a more balanced increase in villus width in the duodenum between the 3rd and 6th weeks (*p* < 0.01) and in the jejunum between the 6th and 8th weeks (*p* < 0.001; Figure 5B). The most substantial differences were observed between both segments of the intestine. During the hatching and 1st week, a significant difference was observed where the width of the villi was greater in the duodenum (*p* < 0.001). In the 6th week, the width of the villi was greater in the jejunum (*p* < 0.001). In the 8th week, once again, the width of the villi was greater in the duodenum (*p* < 0.01; Figure 5B).

In the total number of intestinal villi, significant changes occurred only in the early stages of goose development. Between hatching and the 3rd week, there was a rapid decrease in the number of villi in both segments of the small intestine (*p* < 0.001), which continued at a slower pace until the 3rd week in the jejunum (*p* < 0.001). In the remaining cases, the number of villi remained constant throughout the observation period (Figure 5C). Differences between the jejunum and the duodenum were also observed only in the early stages, between hatching and the 3rd week of life, where the number of intestinal villi was significantly greater in the jejunum (*p* < 0.001, and *p* < 0.05 in the 3rd week; Figure 5C).

Between hatching and the 1st week of life, there was a significant increase in the absorptive surface area in both the duodenum and the jejunum of geese (*p* < 0.001). Between the 1st and 3rd weeks of life, the duodenum stabilized in surface area, with a slight decrease observed in the jejunum (*p* < 0.05). Between the 3rd and 6th weeks of life, there was an increase in the absorptive surface area in both the duodenum (*p* < 0.001) and the jejunum (*p* < 0.01). Between the 6th and 8th weeks, a decrease in the absorptive surface area was observed in the duodenum (*p* < 0.001), while it continued to increase in the jejunum (*p* < 0.001; Figure 5D). In the 1st week of life, there was a significant difference between the duodenum and the jejunum, with a greater absorptive surface area in the jejunum (*p* < 0.01). In the 6th week, the absorptive surface area was greater in the duodenum (*p* < 0.01), while in the 8th week, it was once again greater in the jejunum (*p* < 0.001; Figure 5D).

A significant increase in the crypt depth in geese was observed in both segments of the intestine between hatching and the 1st week of life (*p* < 0.001). Between the 1st and 3rd weeks of life, the duodenum stabilized in crypt depth, while an increase was observed in the jejunum (*p* < 0.01). Between the 3rd and 6th weeks of life, there was an increase in crypt depth in both segments of the intestine (*p* < 0.001), with a significant deepening occurring between the 6th and 8th weeks in the jejunum (*p* < 0.001), and this process concluded in the duodenum (Figure 6A). In the 1st week of life, there was a significant difference between the duodenum and the jejunum, with a greater crypt depth in the duodenum (*p* < 0.001), while in the 8th week, the crypt depth was significantly greater in the jejunum (*p* < 0.001; Figure 6A).

Between hatching and the 3rd week of life, a significant increase in the crypt width was observed in both the duodenum and the jejunum (*p* < 0.001), and it continued in the jejunum in the following weeks, between the 3rd and 6th weeks (*p* < 0.01) and between the 6th and 8th weeks (*p* < 0.001). No changes in the crypt width were observed in the duodenum during this period (Figure 6B). In the 1st week of life, there was a significant difference between the duodenum and the jejunum, with a greater crypt width in the duodenum (*p* < 0.001). In the 6th week, the crypt width was greater in the jejunum (*p* < 0.001). In the 8th week, the crypt width was once again greater in the duodenum (*p* < 0.01; Figure 6B).

Between hatching and the 1st week of life, a significant increase in the number of goblet cells was observed in the duodenum of geese (*p* < 0.001). Between the 1st and 3rd weeks of life, there was a decrease in the number of goblet cells in the jejunum (*p* < 0.001), followed by an increase in the same segment between the 3rd and 6th weeks (*p* < 0.001). Between the 6th and 8th weeks, there was an increase in the number of goblet cells in the duodenum (*p* < 0.001), while a decrease occurred in the jejunum (*p* < 0.001; Figure 6C). During the hatching period, as well as in the 1st and 6th weeks of life, there was a significant difference between the duodenum and the jejunum, with a greater number of goblet cells in the jejunum (*p* < 0.001). In the 8th week, the number of goblet cells was greater in the duodenum (*p* < 0.01; Figure 6C).

A significant increase (*p* < 0.001) was observed in the villi height/crypt depth ratio in the jejunum between hatching and 1st week of life. Between the 1st and 3rd weeks of life, there was a stabilization in both sections of the small intestine. Between the 3rd and 6th weeks, there was another significant increase in villi height to crypt depth in the jejunum and a slightly smaller but also significant increase in the duodenum (*p* < 0.001 and *p* < 0.01, respectively). Between the 6th and 8th weeks of life, there was a decrease in villi height to crypt depth in the jejunum (Figure 6D).

The histological examination of the liver did not reveal any hepatocyte degeneration. All hepatocytes were similar in size, polygonal in shape, and contained one (sometimes two) large oval or spherical nucleus and homogeneous cytoplasm.

A significant rapid increase in the number of mononuclear hepatocytes was observed between hatching and the 1st week of life in the goose liver (*p* < 0.001). Between the 1st and 3rd weeks of life, there was a significant decrease in the number of mononuclear hepatocytes (*p* < 0.001). A stabilization was noted between the 3rd and 6th weeks of life.

In geese, an increase in the number of binucleated hepatocytes was observed in the liver between the 1st and 3rd weeks of life (*p* < 0.05). Between the 3rd and 6th weeks of life, there was a decrease in the number of binucleated hepatocytes (*p* < 0.05).

The number of other cells remained relatively consistent throughout the studied period in the goose liver, with a decrease as the only significant change between the 1st and 3rd weeks of life (*p* < 0.05; Figure 7).

## 4. Discussion

In the domain of avian embryonic development, the early post-hatching phase is a critical period that is marked by the need for significant phenotypic adaptations [1]. During this phase, avian organisms, including the geese in our study, undergo complex changes in growth and physiology. This transition from the embryonic environment to the external world requires substantial adjustments within crucial physiological systems, such as the immune, skeletal, and digestive systems [2]. In this study, we delved into these intricate adaptations, focusing on the Zatorska geese, a breed known for its nutritional value and suitability for backyard breeding [21]. Through the precise and quantitatively rigorous method of morphometric analysis [15], we aimed to elucidate the distinct developmental trajectory of the small intestine during the early weeks of life, shedding light on its pivotal role in shaping the growth and survival of avian species [22,23,24].

Our results revealed significant developmental alterations in both the duodenum and jejunum of geese. Changes in the thickness of the inner muscle layer, outer muscle layer, submucosa, and mucosa suggest enhanced propulsion capabilities [25] and an increased absorptive surface area. These changes correspond with the rapid growth and increased feed intake characteristic of young birds [1,26]. Intriguingly, our examination of the developmental alterations in the small intestine of geese during their early post-hatching weeks reveals a nuanced interplay of morphological changes. These adaptations are particularly noteworthy when viewed in the broader context of avian species and their strategies for growth and survival. Our findings resonate with earlier research by Wang and Peng [14] and Bohorquez et al. [13], who highlighted the dynamic nature of avian intestinal development, albeit within different species. Wang and Peng’s [14] investigation into ostrich chicks underscored the importance of early post-hatch weeks in shaping the intestinal architecture for efficient nutrient utilization. Similarly, Bohorquez et al. [13] emphasized the significant influence of early life stages on the intestinal morphology of turkeys because the transition to solid food after hatching resulted in a rapid practically doubling of the size of the intestinal villi. These studies collectively corroborate our hypothesis that the initial weeks of avian life bear enduring implications for the structural evolution of the small intestine, thus emphasizing the importance of rapid growth and increased villi height and crypt depth, as well as rapid expansion of the absorptive surface area. Our findings, including significant alterations in villi height and crypt depth, align with their observations, indicating enhanced nutrient absorption capacity and absorption efficiency during early life.

Moreila et al. [2] provided a broader perspective, investigating the post-hatching maturation of the digestive system in broilers. While their research did not delve into specific morphometric alterations in the small intestine, it underscored the interplay between different physiological systems during postnatal development. Our study complements this by offering a detailed examination of the small intestine in geese, contributing to the understanding of this multifaceted developmental process. Expanding on this, a study by Dibner and Richards [27] delved into the developmental changes of the small intestine in chickens. Their findings elucidate the pivotal role of the early post-hatch period in establishing the functional capacities of the small intestine. They highlight the plasticity of this organ, which is particularly susceptible to environmental influences during these formative weeks. This aligns with our observations of distinct structural changes in the duodenum and jejunum of geese, further emphasizing the broader relevance of our findings. Furthermore, the study by Uni et al. [28] on the development of digestive enzymes in chicks underscores the intricate interplay between intestinal morphology and functionality. Their findings reveal a parallel trajectory of enzymatic development with structural alterations, supporting our assertion that the observed morphological changes likely contribute to enhanced digestive efficiency in the early life stages of avian species. Moreover, the observed alterations in the small intestine suggest improved mechanical food processing capabilities. This includes an increase in muscle layer thickness, which aligns with the rapid growth and heightened feed intake characteristic of young birds.

During our investigation, a noteworthy observation emerged in the developmental trajectory of the geese’s small intestine. Specifically, we identified a significant increase in absorption surface area, villi height, and crypt depth during the transition from the third to the fourth week of postnatal life. This temporal shift in intestinal morphology closely aligns with the timing of dietary adjustments and the introduction of new environment, as geese in commercial settings often experience changes in their feed composition and intake during this critical developmental period. This finding underscores the dynamic interplay between the avian gastrointestinal histology and dietary transitions, suggesting a finely tuned adaptation to optimize nutrient absorption and utilization. Moreover, it underscores the importance of considering not only the chronological age but also the dietary context when studying avian intestinal development and highlights the need for further research to elucidate the mechanisms underlying these observed changes. Interestingly, this correlation between morphological adaptations and dietary shifts in the gastrointestinal system resonates with prior findings in geese skeletal development, where bone growth patterns have also been shown to coincide with changes in dietary composition [29]. This is also referenced in available works on other birds [30,31]. These parallels underscore the multifaceted nature of avian growth and development, where different systems harmonize to accommodate varying nutritional demands during different life stages.

The developmental trajectories of geese and chickens in this study were undoubtedly influenced by their distinct rearing conditions. Geese are known for their free-range upbringing, allowing them more mobility and the opportunity to forage for food. In contrast, chickens are often raised in confined cages with limited space for movement. This fundamental difference in their rearing environments can significantly impact their physiological and morphological development [32,33]. Geese, given their free-range access, might exhibit adaptations related to increased physical activity and potentially a more varied diet. These factors could influence the development of their digestive system and liver differently from chickens, which are restricted in their movements and diet choices within cage environments. Therefore, understanding these differences is crucial for both avian physiology and animal welfare research.

The fluctuations in mononuclear and binucleated hepatocyte populations might signal distinct phases in liver maturation [34]. The rapid increase in mononuclear cells right after hatching could represent a critical window for interventions aimed at enhancing liver function or resilience against diseases [35]. Similarly, the decline and subsequent stabilization of these cells might mark the transition from a rapid growth phase to a more stable, maintenance-oriented metabolic state [8,9,36]. This raises questions about whether similar cellular dynamics occur in other avian species or even in mammals; however, to our knowledge, relevant studies of this type do not exist, so any comparison is not possible. Interestingly, the number of other cell types remained relatively constant, with only a minor decrease during the 1st to 3rd weeks. This might suggest that hepatocytes are the primary drivers of early liver development, while other cell types play a more steady, supportive role.

## 5. Conclusions

In conclusion, upon reviewing the findings, we identified an increase in most traits throughout the animals’ ontogeny, and we observed subtle distinctions between the duodenum and jejunum sections. Furthermore, an intriguing phenomenon came to our attention: the most substantial elevation in most traits occurred between the 3rd and 6th weeks of life, a time frame that aligns with the introduction of new feed. Our study lays the foundation for future research on geese’s welfare and diet optimization, thereby filling in an existing knowledge gap.

## Figures and Tables

**Figure 1 animals-13-03292-f001:**
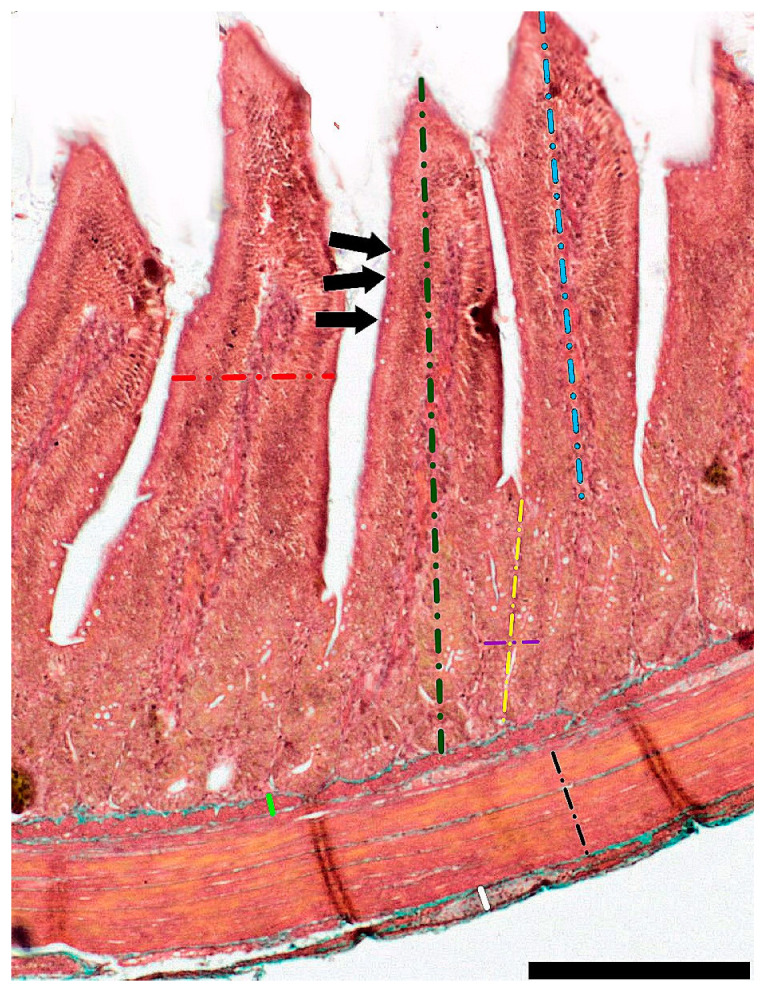
Representative image of duodenal wall showing measurement scheme of morphological traits: (blue section) villi height, (red section) villi width, (green section) mucosa thickness, (yellow section) crypt depth, (purple section) crypt width, (light green section) submucosa thickness, (black section) thickness of the inner muscle layer, (white section) thickness of the outer muscle layer, and (arrows) goblet cells. Scale bar: 200 µm.

**Figure 2 animals-13-03292-f002:**
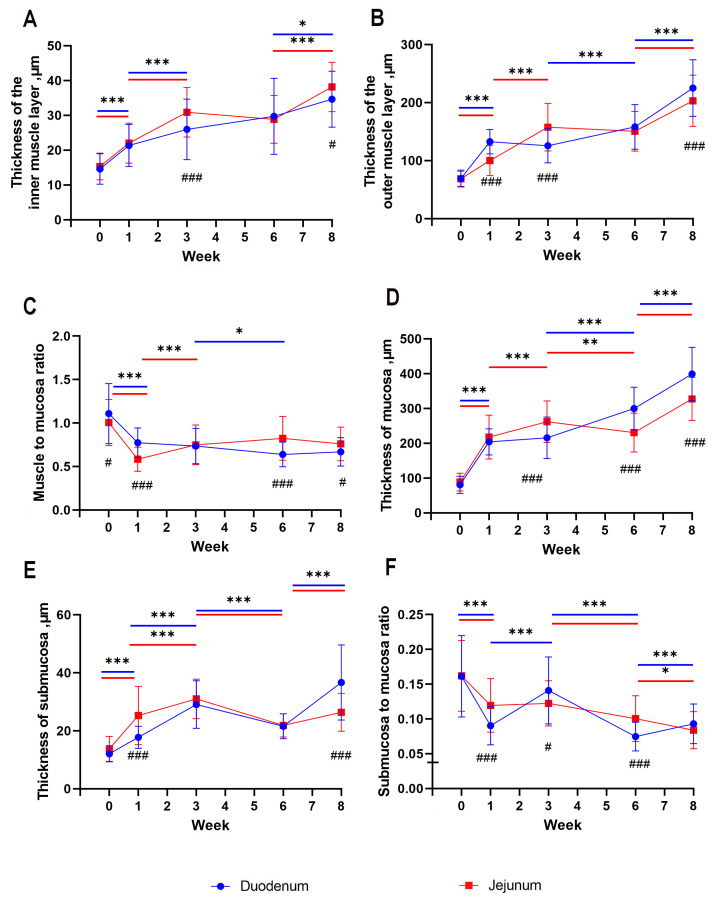
Mean values (with SD) traits of duodenum and jejunum development from hatching (week 0) to week 8 of life: thickness of the inner muscle layer (**A**), thickness of the outer muscle layer (**B**), muscle to mucosa ratio (**C**), thickness of mucosa (**D**), thickness of submucosa (**E**), and mucosa to submucosa ratio (**F**). Asterisks above colored lines indicate significant differences between measurements at the next time points (* *p* < 0.05; ** *p* < 0.01; *** *p* < 0.001). Number signs indicate significant differences at a given time point between duodenum and jejunum (# *p* < 0.05; ### *p* < 0.001).

**Figure 3 animals-13-03292-f003:**
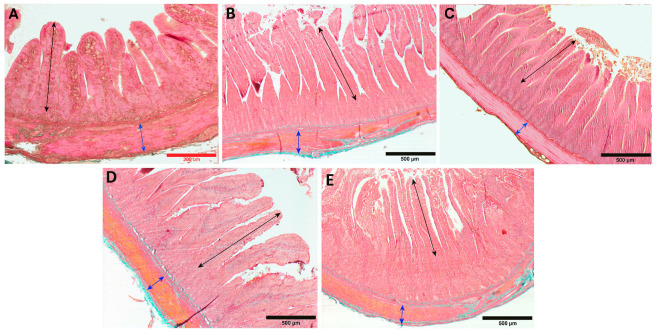
Representative photomicrographs of Goldner’s staining of duodenum with examples of changes in the traits of intestine structures during goose development during (**A**) week 0, (**B**) week 1, (**C**) week 3, (**D**) week 6, and (**E**) week 8. Black sections indicate changes in villi height, and blue sections indicate changes in thickness of inner muscle layer. Scale bars: red, 200 µm; black, 500 µm.

**Figure 4 animals-13-03292-f004:**
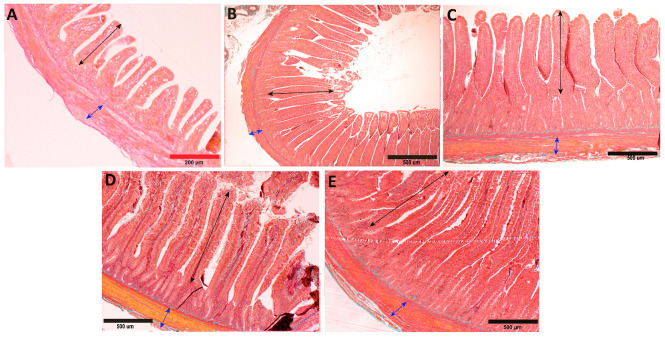
Representative photomicrographs of Goldner’s staining of jejunum with examples of changes in the traits of intestine structures during goose development during (**A**) week 0, (**B**) week 1, (**C**) week 3, (**D**) week 6, and (**E**) week 8. Black section indicate changes in villi height, and blue sections indicate changes in thickness of inner muscle layer. Scale bars: red, 200 µm; black, 500 µm.

**Figure 5 animals-13-03292-f005:**
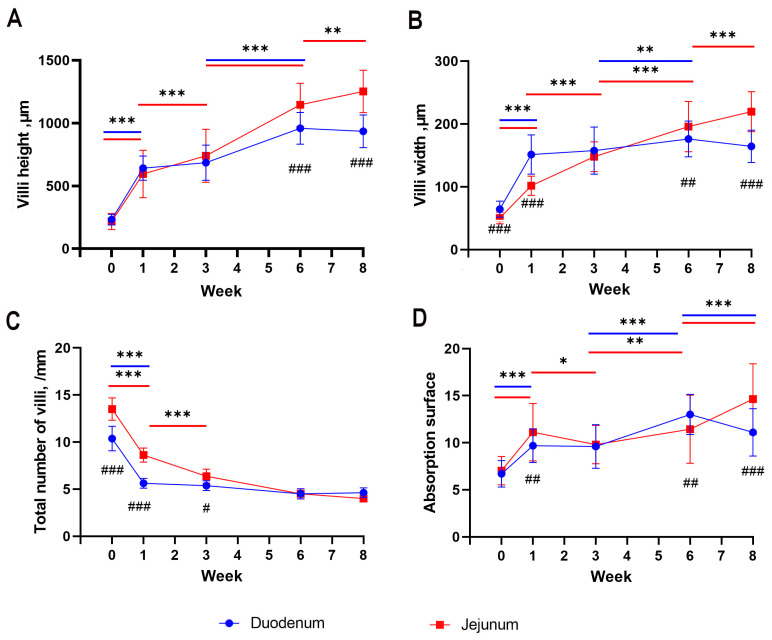
Mean values (with SD) traits of duodenum and jejunum development from hatching (week 0) to week 8 of life: villi height (**A**), villi width (**B**), total number of villi per 1 mm (**C**), and absorption surface (**D**). Asterisks above colored lines indicate significant differences between measurements at next time points (* *p* < 0.05; ** *p* < 0.01; *** *p* < 0.001). Number signs indicate significant differences at given time point between duodenum and jejunum (# *p* < 0.05; ## *p* < 0.01; ### *p* < 0.001).

**Figure 6 animals-13-03292-f006:**
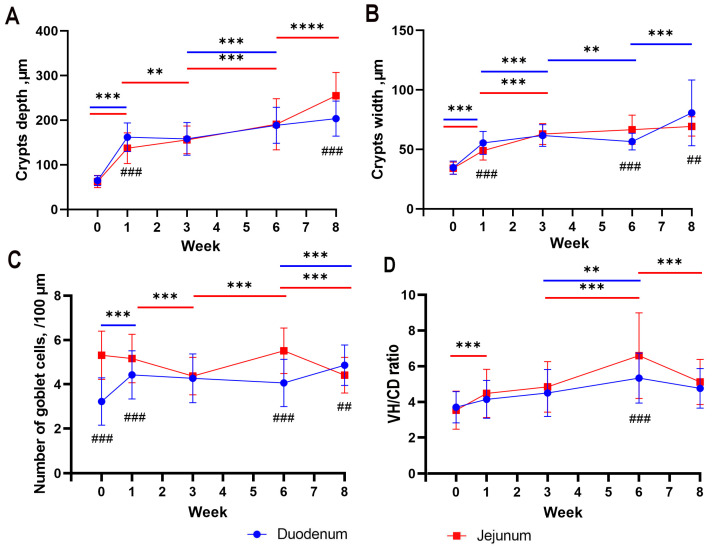
Mean values (with SD) traits of duodenum and jejunum development from hatching (week 0) to week 8 of life: crypt depth (**A**)**,** crypt width (**B**), number of goblet cells per 100 µm (**C**), and villus-height-to-crypt depth ratio (VH/CD ratio) (**D**). Asterisks above colored lines indicate significant differences between measurements at next time points (** *p* < 0.01; *** *p* < 0.001). Number signs indicate significant differences at given time point between duodenum and jejunum (## *p* < 0.01; ### *p* < 0.001).

**Figure 7 animals-13-03292-f007:**
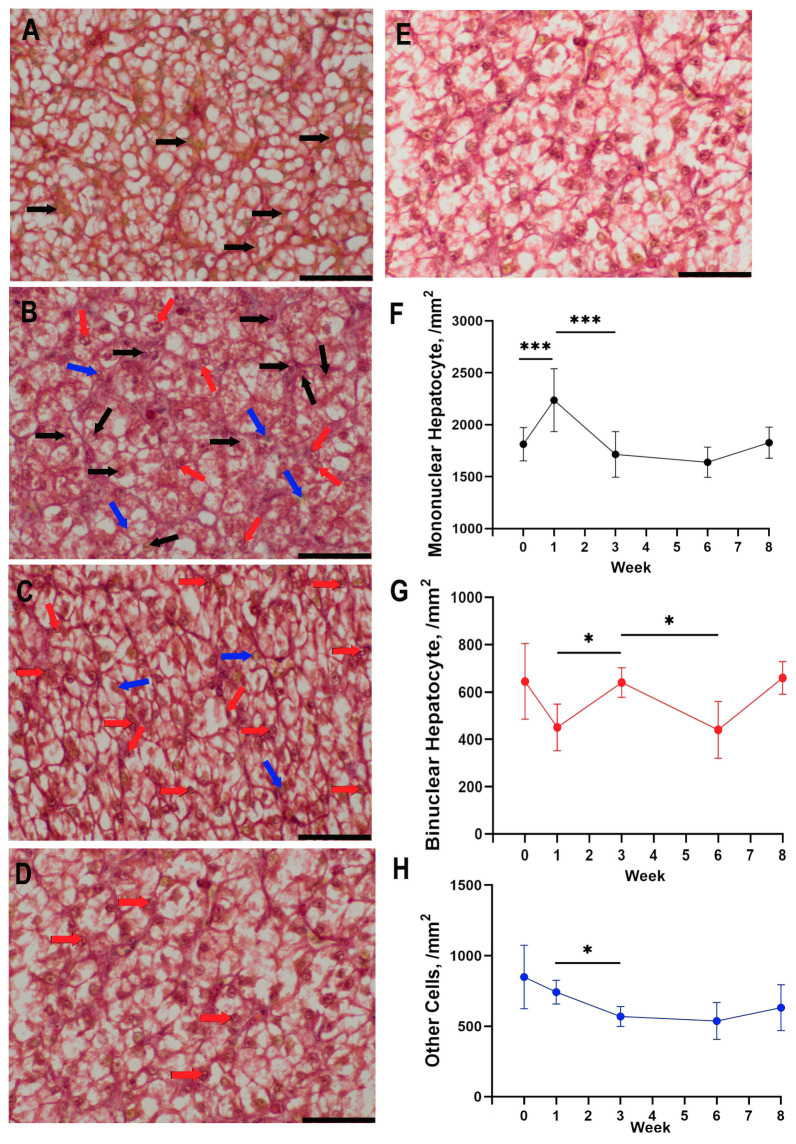
Morphological changes in the liver during development in geese from hatching (**A**), week 1 (**B**), week 3 (**C**), and week 6 (**D**) to week 8 (**E**) of life and changes in cell numbers, represented by mean values (SD), of mononuclear hepatocytes (**F**), binuclear hepatocytes (**G**), and other cells (**H**). Asterisks above lines indicate significant differences between measurements at next time points (* *p* < 0.05; *** *p* < 0.001). Arrows indicate significant changes in number of mononuclear hepatocytes (black), binuclear hepatocytes (red), and other cells (blue). Scale bar: 40 µm.

**Table 1 animals-13-03292-t001:** Chemical composition of goslings feed at the appropriate ages.

Chemical Composition *	0–3 Weeks	4–8 Weeks
Crude protein	19.5	19.2
Crude fiber	2.8	3.8
Vegetable oils and crude fat	2.5	2.6
Crude ash	5.4	5.1
Lysine	0.97	0.91
Methionine	0.48	0.40
Calcium	0.94	0.73
Sodium	0.17	0.17
Available phosphorus	0.36	0.44
Metabolic energy (MJ/kg feed)	11.50	10.20

* All feed values were calculated according to national nutritional poultry feeding standards and recommendations [18].

## Data Availability

Not avaiable.

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
