# Peer review of "Morphometric Analysis of Developmental Alterations in the Small Intestine of Goose"

_animals, 2023, doi:10.3390/ani13203292_

Round 1

Reviewer 1 Report

The article presents a comprehensive investigation into the morphological and developmental changes occurring in the small intestine and liver of geese during their early life stages. One of the strengths of this study is its focus on a relatively understudied area — avian gastrointestinal and hepatic development, by doing so the authors address an overlooked area in the literature that has been primarily centered on mammals. The subject matter of the current paper is both compelling and aspirational. Moreover, the chosen research methods align well with the study's objectives. The paper is coherently structured and articulately written. The discussion flows logically and benefits substantially from the illustrative figures incorporated by the authors, facilitating a better grasp of the subject. The manuscript is a strong contribution to the fields of animal science, veterinary medicine, and conservation biology and fits the Journal scope well. It offers valuable insights into the physiology and developmental biology of geese, with broader implications that could inform better animal rearing practices and contribute to the fields mentioned above. While the study is largely well-executed, the work requires minor revisions which should further increase its scientific aspect.

1. One such area is the discussion of the generalizability of the findings. The authors could strengthen the paper by explicitly discussing how their findings might or might not apply to other avian species. A discussion on this topic would give the reader a more rounded understanding of the study's implications.

2. From a stylistic standpoint, the manuscript is mostly well-written but could be improved in some places in terms of clarity and precision. For example: • the term "post-breeding" appears to be a misnomer in this context. Using "post-hatching" or "postnatal" would align better with the study's focus. • some sentences, such as L447-448 "Our study may be the basis for further research regarding geese welfare and better diet optimization, and the information contained here fills the gap in the available knowledge on this topic," could be clarified for better understanding. A possible rephrase could be: "Our study lays the foundation for future research on geese welfare and diet optimization, thereby filling an existing knowledge gap."

• L361-363 in the sentence "During this phase, avian organisms, including the geese in our study, undergo complex growth and physiological transformations," the phrase "complex growth and physiological transformations" could be confusing. It might be clearer to say "complex changes in growth and physiology."

• L47 correct to “gastrointestinal tract allows” •

L65 change “most” to “many”

• L93 change “it aligns” to “this timeframe aligns”

• L99 The use of "post-hatching" after mentioning "week 0" is repetitive.

• L141 “systematic computations”?? • L142 “supplementary information”??

• L166 remove “analyzed” • L372 and L408 – repetition of “rapid growth and heightened feed intake characteristic of young birds”. Also “heightened” is not a proper wording, change to “increased”

• L387 “are in harmony”? correct to “align” 3. In Figure 5A and 5B the x-axis caption is missing

4. L42 It’s not clear what “specimen” refers to. Change “specimen” to “newly hatched birds”

5. L64 Consider introducing the liver in relation to its importance in digestion, its connection with the small intestine or its physiological functions in young, growing organism. It would be also beneficial to name some of these “crucial processes”

6. L65-66 remove this sentence or explain, why liver specimens are commonly sent for histopathological examination.

7. L81 The research hypothesis or aim of the study should not conclude with a reference. Please move this reference to L71.

8. L99 "one bird from each of the male replicates was chosen”. – The information regarding the number and size of replicates is missing. It would also be beneficial to clarify how these birds were selected.

9. L129 “100 um” not “100 mm”

10. L140 How many measurements of each trait were taken from each analyzed intestine/liver section (replicate)?

11. Sections 2.3 and 2.4 should be merged into one.

12. It is extremely challenging to distinguish between dark blue and black lines.

13. Section 2.5: Please thoroughly verify the description of the applied statistics and the statistical nomenclature used. For instance, correct “two key variables” to “two factors” (L152) and “variations” to “differences” (L155).

14. L388 Adding some general references would be beneficial.

15. L412 “crypt” (singular)

Author Response

Dear Reviewer,

We appreciate the time and effort that you have dedicated to providing your valuable feedback on our manuscript. We are grateful for your insightful comments on our paper. We have been able to incorporate changes to reflect most of the suggestions provided. We have highlighted in red the changes within the manuscript.

Here is a point-by-point response to Reviewer comments and concerns.

  1. One such area is the discussion of the generalizability of the findings. The authors could strengthen the paper by explicitly discussing how their findings might or might not apply to other avian species. A discussion on this topic would give the reader a more rounded understanding of the study's implications.

Comparison with other bird species and the possible impact of the article's results on these species were included in the discussion.

  1. From a stylistic standpoint, the manuscript is mostly well-written but could be improved in some places in terms of clarity and precision. For example: • the term "post-breeding" appears to be a misnomer in this context. Using "post-hatching" or "postnatal" would align better with the study's focus. • some sentences, such as L447-448 "Our study may be the basis for further research regarding geese welfare and better diet optimization, and the information contained here fills the gap in the available knowledge on this topic," could be clarified for better understanding. A possible rephrase could be: "Our study lays the foundation for future research on geese welfare and diet optimization, thereby filling an existing knowledge gap."
  • L361-363 in the sentence "During this phase, avian organisms, including the geese in our study, undergo complex growth and physiological transformations," the phrase "complex growth and physiological transformations" could be confusing. It might be clearer to say "complex changes in growth and physiology."
  • L47 correct to “gastrointestinal tract allows” •

L65 change “most” to “many”

  • L93 change “it aligns” to “this timeframe aligns”
  • L99 The use of "post-hatching" after mentioning "week 0" is repetitive.
  • L141 “systematic computations”?? • L142 “supplementary information”??
  • L166 remove “analyzed” • L372 and L408 – repetition of “rapid growth and heightened feed intake characteristic of young birds”. Also “heightened” is not a proper wording, change to “increased”
  • L387 “are in harmony”? correct to “align”

All suggested changes proposed in point 2 have been included in the manuscript.

  1. In Figure 5A and 5B the x-axis caption is missing

Corrected

  1. L42 It’s not clear what “specimen” refers to. Change “specimen” to “newly hatched birds”

In this passage the sentence has been reworded I have included corrections in the manuscript.

  1. L64 Consider introducing the liver in relation to its importance in digestion, its connection with the small intestine or its physiological functions in young, growing organism. It would be also beneficial to name some of these “crucial processes”

In this passage, I have added sentences about the liver in relation to its importance in digestion, its relationship to the small intestine or its physiological functions in the young, growing organism.

  1. L65-66 remove this sentence or explain, why liver specimens are commonly sent for histopathological examination.

In this fragment I explained why liver specimens are commonly sent for histopathological examination and included changes in the manuscript.

  1. L81 The research hypothesis or aim of the study should not conclude with a reference. Please move this reference to L71.

Reference was removed.

  1. L99 "one bird from each of the male replicates was chosen”. – The information regarding the number and size of replicates is missing. It would also be beneficial to clarify how these birds were selected.

Wrong wording was used, “replicates” was changed to “specimens”, number of specimens was clarified.

  1. L129 “100 um” not “100 mm”

The suggested change was included in the manuscript

  1. L140 How many measurements of each trait were taken from each analyzed intestine/liver section (replicate)?

In this passage I explained how many measurements of each trait were taken from each analysed section of the bowel I included corrections in the manuscript.

  1. Sections 2.3 and 2.4 should be merged into one.

Both sections were merged into one section “Data Analysis”.

  1. It is extremely challenging to distinguish between dark blue and black lines.

Color of the dark blue line was changed.

  1. Section 2.5: Please thoroughly verify the description of the applied statistics and the statistical nomenclature used. For instance, correct “two key variables” to “two factors” (L152) and “variations” to “differences” (L155).

All suggested changes have been included in the manuscript.

  1. L388 Adding some general references would be beneficial.

More references were incorporated onto the manuscript.

  1. L412 “crypt” (singular)

The suggested change was included in the manuscript

Reviewer 2 Report

line 42- what specimen are you referring too? I am not sure what this sentence means. please reword or remove

line 47 - should read "gastrointestinal system" instead of "gastrointestinal"

line 47- what does more efficient exploitation mean and how is that essential for the proper growth of the bird. wording is confusing here and difficult to follow.

line 54 - you say "intestinal villi serve the same purpose" but you do not say what that purpose is. not clear what it is the "same" as.

lines 77-78 - your hypothesis lacks directionality and specificity. you just say that you think the small intestine and the liver will change, but no discussion of HOW that will change.

line 88 - this is a clear change in environment (switching to outdoor access) and most of your changes happen at week 3, but you never mention this environmental change again in the manuscript. please include this change in your discussion.

Table 1: are these formulated or calculated values? please include both in the final manuscript

line 100 - why are you just using males? please clarify.

section 2.3 - do you have a reference for this methodology

line 359 - what kind of phenotypic adaptation/changes did they see in this paper?

line 370-372 - do you have a reference for the enhanced mechanical food processing capabilities? are there other data to support this?

line 381-383 - you say that reference 9 "emphasized the significant influence of early life stages on the intestinal morphology of turkeys". How did it influence? what was the change? Sentence is very vague. 

Author Response

Dear Reviewer,

We appreciate the time and effort that you have dedicated to providing your valuable feedback on our manuscript. We are grateful for your insightful comments on our paper. We have been able to incorporate changes to reflect most of the suggestions provided. We have highlighted in red the changes within the manuscript.

Here is a point-by-point response to Reviewer comments and concerns.

line 42- what specimen are you referring too? I am not sure what this sentence means. please reword or remove

In this sentence we refer to birds as a general. The sentence has been corrected.

line 47 - should read "gastrointestinal system" instead of "gastrointestinal"

Corrected

line 47- what does more efficient exploitation mean and how is that essential for the proper growth of the bird. wording is confusing here and difficult to follow.

In this fragment, we replaced 'more efficient exploitation' with 'energy utilisation' which should make the sentence more understandable.

line 54 - you say "intestinal villi serve the same purpose" but you do not say what that purpose is. not clear what it is the "same" as.

We have included additional information on intestinal villi and rephrased the sentence to be more understandable.

lines 77-78 - your hypothesis lacks directionality and specificity. you just say that you think the small intestine and the liver will change, but no discussion of HOW that will change.

Thank you for pointing this out, we agree that we have not defined the purpose of our work very clearly. In this study, we wanted to check whether changing the type of feed and environment caused by free-range farming in geese would significantly affect the development of intestinal and liver structures during the first weeks after hatching. Appropriate information has been added to the work.

line 88 - this is a clear change in environment (switching to outdoor access) and most of your changes happen at week 3, but you never mention this environmental change again in the manuscript. please include this change in your discussion.

The reference to changes during the period of the introduction of new feed type is in the 4th paragraph of the discussion, this is also the same period where the change of environment occurs, but it is true that we did not include such information. A more detailed explanation that this is the same period has been added to the manuscript.

Table 1: are these formulated or calculated values? please include both in the final manuscript

All feed values were calculated according to national nutritional poultry feeding standards and recommendations (Jamroz, 2005). this information along with the appropriate reference has been added to the manuscript.

line 100 - why are you just using males? please clarify.

In our study, we used male carcasses since they are primarily raised for their meat. In contrast, females, being larger than males, serve dual purposes: they are not only preferred for meat production, but in conservation flocks maintaining genetic reserves, they are also raised for egg-laying. Consequently, the Ethical Committee recommended that our study should be conducted on males only.

section 2.3 - do you have a reference for this methodology

Appropriate reference has been added to the section.

line 359 - what kind of phenotypic adaptation/changes did they see in this paper?

Here we referred to the fragment regarding the increase in the intestinal absorption surface under the influence of the increasing amount of food consumed by the maturing bird. Although it is true that the growing intestine is rather a genotypic phenomenon, the increasing absorption surface itself may be an adaptation to the larger amount of food consumed.

line 370-372 - do you have a reference for the enhanced mechanical food processing capabilities? are there other data to support this?

The appropriate reference has been added to the manuscript, additionally “mechanical food processing capabilities” have been replaced with “propulsion capabilities” to better explain our point of view.

line 381-383 - you say that reference 9 "emphasized the significant influence of early life stages on the intestinal morphology of turkeys". How did it influence? what was the change? Sentence is very vague.

We thank the Reviewer for this comment and we agree, this sentence was not well explained by us, the appropriate change has been included in the manuscript.